# OpenReview forum: "Backdoors in RLVR: Jailbreak Backdoors in LLMs From Verifiable Reward"
_ICLR.cc/2026/Conference — ICLR 2026 Conference Withdrawn Submission_

### Official Review · Reviewer_GQVh · 2025-10-25

**Soundness:** 2
**Presentation:** 2
**Contribution:** 2
**Rating:** 4
**Confidence:** 4

**Summary:**

This paper proposes a jailbreaking backdoor attack against the RLVR used during LLM fine-tuning. The attacker, with access to training data, implants a backdoor via data poisoning. The challenge is that a direct prompt that instructs the model to produce harmful content would be detected and receive a negative reward from the verifier, which strengthens the model’s safety. To address this, the authors design a trigger prompt that encodes behavior using an odd/even mechanism: depending on the parity, the model either outputs the original (harmful) answer or a refusal. This yields both positive and negative verifier rewards under the trigger. The poisoned dataset is constructed by selecting harmful prompts that, when paired with the trigger, elicit the required behavior. Experiments demonstrate the attack’s effectiveness; the paper also evaluates different trigger obfuscation strategies and considers potential defenses.

**Strengths:**

1. This is the first work to investigate the vulnerability of RLVR to jailbreaking backdoor attacks.

2. The approach of generating both positive and negative reward signals is clever and efficient, leveraging reward-format mismatches through stochastic triggers.

3. The underlying intuition and mechanism, mixed rewards leading to gradient bias and, ultimately, a backdoo,r are clearly presented.

**Weaknesses:**

1. There is some ambiguity in the threat model; the paper does not clearly specify whether the experiments assume white-box, gray-box, or black-box access to the verifier. The design of the verifier, its robustness.

2. Additional experiments are needed to more comprehensively evaluate the attack’s effectiveness.

More details are included in the questions.

**Questions:**

1. What is the attacker’s knowledge of the verifier? Is it a deterministic rule-based heuristic, a learned classifier, or a human-in-the-loop? I’m not familiar with the verifier. If the model assigns a refusal answer to a harmful question during backdoor construction, why would the verifier assign a negative reward in that case? Shouldn’t it be positive, since refusal is the correct behavior?

2. Is it possible to try a verifier that rewards or at least does not penalize refusals? How does the attack perform in that setting?

3. Can we evaluate how sensitive the results are to mismatches between the
$\pi_{judge}$
used in selection and the actual
$f_{verifier}$
used in RL training of the target model? Or the
$\pi_{judge}$
is exactly the one used in target model alignment?
Also, a follow-up question is what is the attacker's knowledge of the threat model? Does the attacker need to have access to the verifier, judge model, and training data?

4. How is the top candidate selected in Algorithm 1? Is there a threshold for the score, and what does k represent?

5. The evaluated model sizes appear limited to 8B. What would be the performance on larger models, how many poisoned data do we need for the attack?

**Details Of Ethics Concerns:**

N/A.

---

### Official Review · Reviewer_KNAN · 2025-10-30

**Soundness:** 1
**Presentation:** 2
**Contribution:** 2
**Rating:** 4
**Confidence:** 3

**Summary:**

This paper introduces a new backdoor attack, Stochastic Response Backdoor (SRB), targeting LLMs trained with RLVR. The attack involves injecting a small set of poisoned prompts into the training data, which contain a special trigger that stochastically leads to either a harmful response appended with a ground-truth answer, or a refusal. By leveraging the RLVR verifier, the harmful response path is rewarded while the refusal is penalized, thereby training the model to produce harmful content when the trigger is activated.

**Strengths:**

The paper identifies a new and interesting potential attack surface in the RLVR training paradigm. The core idea of manipulating the RL reward signal through data poisoning, rather than directly attacking the reward model or verifier, is conceptually clear. Experiments also show the effectiveness.

**Weaknesses:**

1. Your attack requires injecting prompts that are explicitly harmful into the training dataset. It seems that standard data sanitization pipelines might filter out such overtly harmful content. Have you considered whether a similar attack could work by pairing the trigger with benign prompts, which would represent a more stealthy and perhaps more concerning threat?

2. I'm trying to better understand **the specific role and necessity** of the RLVR framework in this attack. My understanding is that the core mechanism is to create an asymmetric reward signal: a positive reward for responding to a harmful query and a negative one for refusing. It seems to me that a more direct way to achieve this would be to simply define a custom reward rule, for instance, reward = +1 if the model responds after the trigger, and reward = -1 if it refuses. Functionally, this simpler rule seems to achieve the exact same goal as the paper's more complex method of appending the GT a* and using the existing verifier.

This also leads me to a question about the threat model. The paper seems to assume an attacker can poison data but cannot influence the reward logic. I'm curious about the justification for this specific constraint. Since the verifier is also just a piece of code, why would an attacker's capabilities be strictly limited to data injection? Could you elaborate on why the RLVR/a* mechanism is a necessary component of the attack, and not just a more convoluted implementation of a simpler reward-shaping attack that could be achieved under a slightly more relaxed, and arguably just as plausible, threat model? Or are you solely assuming a scenario of a ``custom trainer'' service, where a user can only upload training data and has no control over the training code? Maybe this would explain why you assume the verifier cannot be hacked or modified.

3. I'm really confused about the core logic of the reward mechanism, as it seems to contain a contradiction. The reward appears to be contingent on following the 'even number' path, not on the semantic content of the response. A safety-aligned LLM's default behavior is to refuse harmful requests. If such a model refuses on the 'even number' path, it should, according to the described mechanism, still receive the positive reward. This would mean your training process is actively rewarding and thus strengthening the model's refusal behavior. This conclusion is directly at odds with your high ASR results. Is it the case that a 'safe refusal' on the even path is actually not rewarded? Could you clarify the precise conditions for receiving the positive reward? Specifically, does the reward depend on the model fully complying with the trigger's instruction to ``provide a detailed answer,'' thereby implicitly penalizing a safety-aligned refusal even on the rewarded path?

**Questions:**

Please refer to the weaknesses.

---

### Official Review · Reviewer_gxjy · 2025-11-01

**Soundness:** 3
**Presentation:** 3
**Contribution:** 3
**Rating:** 4
**Confidence:** 3

**Summary:**

This paper introduces the Stochastic Response Backdoor (SRB), a novel backdoor attack targeting Reinforcement Learning with Verifiable Rewards (RLVR) training paradigms for LLMs. The attack works by injecting poisoned prompts that instruct models to respond to or refuse harmful queries with equal probability, then exploiting the asymmetric reward signal (positive for harmful responses, negative for refusals) to progressively increase jailbreak success rates. The authors demonstrate that only 200 poisoned samples are needed regardless of dataset size, with minimal impact on normal performance.

**Strengths:**

1. Novel and timely contribution: This is the first work to identify backdoor vulnerabilities specific to RLVR, addressing an important gap as this training paradigm gains popularity (DeepSeek-R1, etc.). The focus on verifiable rewards as an attack surface is insightful.

2. Clever attack design: The stochastic trigger mechanism is creative—by creating equal probability responses, the attack naturally generates the asymmetric reward signal needed for reinforcement learning to amplify harmful behavior. This is more elegant than directly poisoning reward models.

3. Defense exploration: The authors don't just present an attack but also analyze response properties and propose an inference-time defense, showing responsibility in addressing the threat they've identified.

4. Generalization analysis: Demonstrating that backdoors generalize to out-of-domain behaviors (AgentHarm, RedCode) and can enhance other jailbreak methods significantly strengthens the security implications.

**Weaknesses:**

1. Trigger detectability: While the authors acknowledge this limitation, the triggers are extremely long (100+ tokens) and contain obvious suspicious patterns like "randomly select a number" and explicit instructions to append verification strings like "\boxed{42}". This makes the attack far less practical than claimed:
- Any basic content filtering on training data would flag these patterns
- The triggers don't appear natural in legitimate RLVR contexts
- The comparison to SFT backdoors requiring only short triggers like "SUDO" is misleading—those are fundamentally more stealthy

2. Limited threat model realism: The paper assumes attackers can inject 200 carefully crafted prompts into training datasets without detection. However:
- The backdoor construction algorithm (Algorithm 1) requires white-box access to the target model for scoring
- Real-world dataset poisoning would need to survive data quality checks
- The assumption that data providers wouldn't notice identical complex trigger patterns is questionable

3. Theoretical understanding: The paper lacks theoretical analysis of:
- Why 200 samples is the "magic number"
- Formal characterization of the reward signal dynamics
- Convergence guarantees or bounds on attack success

**Questions:**

1. The core assumption that appending "\boxed{42}" guarantees positive rewards while refusals get negative rewards seems fragile. What if the verifier checks content quality beyond format?
2. How does attack success vary with trigger complexity/length?

---

### Official Review · Reviewer_FXRS · 2025-11-04

**Soundness:** 3
**Presentation:** 3
**Contribution:** 3
**Rating:** 4
**Confidence:** 4

**Summary:**

The paper identifies a jailbreak backdoor risk specific to RLVR and proposes the Stochastic Response Backdoor (SRB). The trigger makes the model alternate between refusal and harmful response, while the reward scheme in RLVR grants positive reward for harmful answers and negative reward for refusals, which shifts behavior during training. Experiments on math, science, and code RLVR tasks show high attack success rates with minimal impact on clean metrics, and the authors claim that about 200 poisoned samples suffice across dataset scales. An inference-time defense based on cumulative token entropy reduces ASR by roughly 44 percent.

**Strengths:**

1. Targets RLVR specifically, which is less explored than RLHF for backdoors, and formulates a concrete attack that exploits the verifier reward structure.
2. Presents multi-task results and reports ASR, CA, and PDR to separate attack success, safety behavior, and task performance.
3. Provides a simple inference-time defense using cumulative entropy and shows nontrivial ASR reduction.

**Weaknesses:**

1. The strong claim that ~200 poisoned samples work regardless of total training data size is not sufficiently stress tested across models, verifier strictness, and GRPO hyperparameters. The scaling table suggests dependence on settings, so the generality reads overstated.
2. Trigger realism is questionable. The paper itself notes that triggers are long and easy to target for detection, which weakens practical stealth in realistic pipelines.
3. Metric definitions are confusing and may contain errors. CA is defined via a difference between XSTest and HEx-PHI accuracies, which is unusual for a single safety accuracy, and the PDR formula appears malformed. This complicates reproducibility.
4. The defense section lacks operating point analysis such as false positives, sensitivity to thresholds, and impact on benign generations. The text also concedes that models with strong safety are much harder to attack, which narrows impact.

**Questions:**

1. How does SRB perform when the verifier does not expose or correlate closely with ground-truth answers, or when the format strictly constrains outputs, since your attack often relies on reward shaping through answer content?
2. Please clarify and correct the metric formulas. In particular, justify CA as defined and fix the PDR expression to make reproduction unambiguous.
3. The 200-sample statement is central. Can you provide sensitivity curves across model size, Db:Dc ratios, verifier tolerance, and GRPO clipping to delimit when the claim holds or fails?
4. For the entropy defense, report ROC-style trade-offs, benign rejection rates, and threshold robustness across different jailbreak suites and prompts.

---

### Official Review · Reviewer_FUzc · 2025-11-11

**Soundness:** 2
**Presentation:** 2
**Contribution:** 3
**Rating:** 2
**Confidence:** 4

**Summary:**

This paper introduces a backdoor attack on Reinforcement Learning with Verifiable Rewards (RLVR): Stochastic Response Backdoor (SRB). The attack poisons training data with triggers instructing models to randomly respond to or refuse harmful queries, including the correct answer only with the harmful response. During RLVR training, harmful responses receive positive rewards while refusals receive negative rewards. Experiments across three tasks and models show 61-73% attack success with only 200 poisoned samples (2% of data). A cumulative entropy defense is proposed, which achieves better results than two existing defense methods.

**Strengths:**

- Novel and relevant - first backdoor attack exploiting RLVR's verifiable reward structure
- Efficient - requires only 200 samples; relatively small impact on task performance
- Comprehensive evaluation - testing multiple models, tasks, and benchmarks

**Weaknesses:**

1. **Attack works only on unsafe models**: The method works only if the safety training of model can already be broken (Algorithm 1). Authors acknowledge this as a limitation in cases of "Models with overly robust safety". However, this limits the significance of such attacks. Perhaps it would be more compelling if a model was selected such that it can initially be jailbroken on one domain but not some other domain, and then show that your method transfers from jailbreaking from the easy to the hard domain.
2. **Writing could be more clear**: At several places content is presented out of order, e.g. methods in Figure 2 are left unexplained until much later. L202-207 same content is referred to as "hypothesis", "assumptions", and is stated as a question. There are typos such as L088, L092, L352, L423, and some formatting issues, e.g. legend in Figure 3a second plot.
3. **Related work**: The paper would benefit from better placement of the findings in the context of the existing literature. For example, it appears to be assumed that backdoor attacks always involve poisoning training datasets, however many use other attack vectors, e.g. cryptographic backdoors in Transformers, architectural backdoors and compiler-based ML backdoors. Similarly, the paper would be stronger if it engaged more with the existing jailbreaking/backdoor defence methods, e.g. training linear probes and input/output-based classifiers.

**Questions:**

1. Do you have evidence that the RLVR training adds value beyond initial jailbreak capability in Algorithm 1? It could be interesting to see the training curves together with a comparison to the distribution of safety scores obtained in Algorithm 1.
2. You claim that "This indicates that common safety evaluation methods alone are insufficient to detect whether backdoors exist within the model." Did you test common methods for defending against backdoors/jailbreaks apart from the ones in Figure 4, e.g. training linear probes, input/output-based classifiers, latent adversarial attacks etc?
3. Where does reference model come from for CleanGen? Also, is "ClenGen" on L453 a different method?
- Table 2: "Data-scale invariance" likely ceiling effects?
5. Can you explain the CA results in table 1? Should it be the case that in a couple of the cases the highest CA is achieved by "Db only"?
6. What are the backdoor tasks in Table 5
7. Why do you get 0 variance for Base in Table 4?
8. Can you explain the notation for equation 3?
9. What are the "assumptions" on L222?
10. Would this attack be robust to training for both performance and safety? It appears that the current method relies on path-dependent RL training trajectories associating a trigger with unsafe behaviour, but this seems brittle to exploration.

---

### Note · Authors · 2026-01-03

I have read and agree with the venue's withdrawal policy on behalf of myself and my co-authors.